# Impact of point-of-care panel tests in ambulatory care: a systematic review and meta-analysis

Clare Goyder [ORCID],[1] Pui San Tan,[1] Jan Verbakel,[1,2] Thanusha Ananthakumar,[1] Joseph J Lee,[1] Gail Hayward [ORCID],[1] Philip J Turner,[1] Ann Van Den Bruel[2]

[1]Nuffield Department of Primary Care Health Sciences, University of Oxford, Oxford, UK
[2]Department of Public Health and Primary Care, KU Leuven, Leuven, Belgium

**Correspondence to**
Dr Clare Goyder;
clare.goyder@phc.ox.ac.uk

## ABSTRACT

**Objectives** This article summarises all the available evidence on the impact of introducing blood-based point-of-care panel testing (POCT) in ambulatory care on patient outcomes and healthcare processes.

**Design** Systematic review and meta-analysis of randomised-controlled trials and before-after studies.

**Data sources** Ovid Medline, Embase, Cochrane Database of Systematic Reviews, Cochrane CENTRAL, Database of Abstracts of Reviews and Effects, Science Citation Index from inception to 22 October 2019.

**Eligibility criteria for selecting studies** Included studies were based in ambulatory care and compared POCT with laboratory testing. The primary outcome was the time to decision regarding disposition that is, admission/referral termed disposition decision (DD) time. Secondary outcomes included length of stay (LOS) at the ambulatory care unit/practice and mortality.

**Results** 19 562 patients from nine studies were included in the review, eight of these were randomised-controlled trials, and one was a before-after study. All the studies were based in either emergency departments or the ambulance service; no studies were from primary care settings. General panel tests performed at the POCT resulted in DDs being made 40 min faster (95% CI −42.2 to −36.6, $I^2$=0%) compared with the group receiving usual care, including central laboratory testing. This in turn resulted in a reduction in LOS for patients who were subsequently discharged by 34 min (95% CI −63.7 to −5.16). No significant difference in mortality was reported.

**Discussion** Although statistical and clinical heterogeneity is evident and only a small number of studies were included in the meta-analysis, our results suggest that POCTs might lead to faster discharge decisions. Future research should be performed in primary care and identify how POCTs can contribute meaningful changes to patient care rather than focusing on healthcare processes.

**PROSPERO registration number** CRD42016035426.

## INTRODUCTION
### Background

All ambulatory care physicians frequently encounter diagnostic uncertainties in day-to-day practice. This can lead to missed opportunities for diagnoses or inappropriate referrals to secondary care. Patients with vague or non-specific symptoms can be

### Strengths and limitations of this study

► This study was conducted robustly, following Preferred Reporting Items for Systematic Reviews and Meta-Analyses guidelines using a comprehensive search strategy in the major medical databases, and selection and quality assessment performed by two independent reviewers.

► It offers results on the impact of panel point-of-care tests (POCTs), rather than just their accuracy.

► Included studies were relatively small, and most notably for mortality, may be underpowered to detect clinically relevant differences between laboratory and POCTs.

► Statistical and clinical heterogeneity is evident within our meta-analysis, however the meta-analysis was considered carefully and heterogeneity reduced where possible by ensuring that studies of cardiac and general panel tests were not combined; moreover, we did not combine the before-after study (which also had a high risk of bias) with randomised-controlled trials.

► Four studies excluded critically ill patients and patients with myocardial infarction, which may have biased mortality data.

the most challenging populations to assess.[1] Currently, most ambulatory care units use whole blood tests that are normally transported to and processed by a centralised clinical laboratory.

The technology behind in vitro point-of-care testing (POCT) has developed extensively and the accuracy compared with standard methods for some tests is now established.[2 3] POCT now offers an alternative to conventional laboratory methods; it is performed on site, normally at the bedside and has a short turnaround time of typically 5–15 min.[4] POCT is being employed in a wide variety of healthcare settings and its use is predicted to expand dramatically.[5] Indeed, NHS England have stated that POCTs will be available in urgent treatment centres in the UK from 2019.[6]

## Importance

The use of POCT in ambulatory care has the potential to reduce diagnostic uncertainty and delay and physicians report that they would like to use these tests more, particularly to aid in the diagnosis of acute conditions.[7–9] It is expected that POCT facilitates healthcare processes such as the speed of discharge, leading to better use of healthcare resources or enables quicker diagnosis and referral of patients with serious illness, which may lead to better patient outcomes. Panel tests are especially appealing in this patient group as they test multiple parameters simultaneously from the same finger prick of blood using the same platform, covering a range of conditions frequently found to cause acute presentations to ambulatory care. However, there are potential disadvantages associated with their implementation[10] and little is understood about the impact of POCT panels on day-to-day practice. Thus far, what is lacking is an up-to-date summary of all the available evidence on the impact of blood-based POCT panel in ambulatory care.

## Objective

In this study, we performed a systematic review and meta-analysis to evaluate the quantitative impact of POCT in ambulatory care with a focus on blood-based panel tests.

## Methods

This protocol has been developed according to recommendations from the Cochrane Collaboration[11] and guidelines from the Preferred Reporting Items for Systematic Reviews and Meta-Analyses (PRISMA) statement[12] have been followed.

This systematic review forms part of a series of analyses from a larger overall review (in progress) which will assess the overall quantitative impact of all POCTs in ambulatory care, further subgroup analyses on C-reactive protein (CRP)[13] and influenza[14] have already been published.

## Patient and public involvement

We consulted with an existing patient and public involvement panel of the NIHR Diagnostic Evidence Co-operative (DEC) Oxford specialising in research on in vitro diagnostic technology, who have been involved in a number of previous projects which incorporated POCT. They felt that this systematic review would be very important in defining the evidence for and against use of POCT in ambulatory care. One member described her experience as a patient in another European country where POCT for certain conditions was seen as part of standard care, and her surprise that this was not the case in the UK. They were specifically interested in the potential implications of POCT for facilitating earlier discharge from hospital.

## Search strategy

We searched Ovid Medline (1946 to 2017), Embase (1974 to 2017), Cochrane Database of Systematic Reviews, Cochrane CENTRAL, Database of Abstracts of Reviews and Effects (DARE) and Science Citation Index (1945–present) from inception. This systematic review forms part of a series of analyses from a larger overall review (in progress) which will assess the overall quantitative impact of all POCTs in ambulatory care. This main search was originally performed on 19 November 2015 and then updated on 21 March 2017, following this subgroup analyses on CRP[13] and influenza[14] were published. Studies based in resource poor settings form another subgroup analyses from the overall review and will also be published separately. A further update was performed on 22 October 2019 and was screened to identify papers that assessed the impact of blood-based panel tests in ambulatory care. We did not identify any new studies from this update, the PRISMA diagram in figure 1 summarises the process. A snowballing strategy was used to ensure that the search was as comprehensive as possible. We did not add a study design filter nor apply a language restriction. We performed citation searches of all full-text papers included in final review. The full strategy is included in online supplementary appendix 1.

## Selection of studies

We included randomised-controlled trials (RCTs), non-randomised but experimental and controlled studies including before-after studies. Included studies provided quantitative comparisons of the impact of blood-based POCTs with laboratory testing and were based in ambulatory care.

Screening was divided between seven authors (CG, PST, JV, TA, JL, PT and AVDB); two of these authors independently assessed the potential relevance of all titles and abstracts identified from the electronic search. Full-text papers of all potentially relevant papers were obtained and these were then further assessed by two of the authors. Conflicts were resolved by seeking the opinion of a third author and disagreements were discussed with the team to obtain consensus. The reason for excluding studies was recorded. The most common reasons for exclusion were studies that only focused on diagnostic accuracy and did not consider impact. Another common reason was studies that only included qualitative comparisons or, if they did have quantitative data that did not provide results on both the intervention and the control groups. We also excluded studies that evaluated POCT exclusively for monitoring purposes. We excluded panel tests that were not based on blood samples. Studies that included panel tests as part of a multifaceted intervention or combined blood-based panel tests with single tests or urine tests were also excluded. Systematic reviews were excluded with reference lists checked for potentially relevant studies for inclusion. For this subgroup analysis, appropriate studies were selected independently by two researchers.

## Data extraction and quality assessment

Data extraction was performed by one author and independently checked by a second author. The authors extracted the following data from included studies: general study information (authors, title, publication

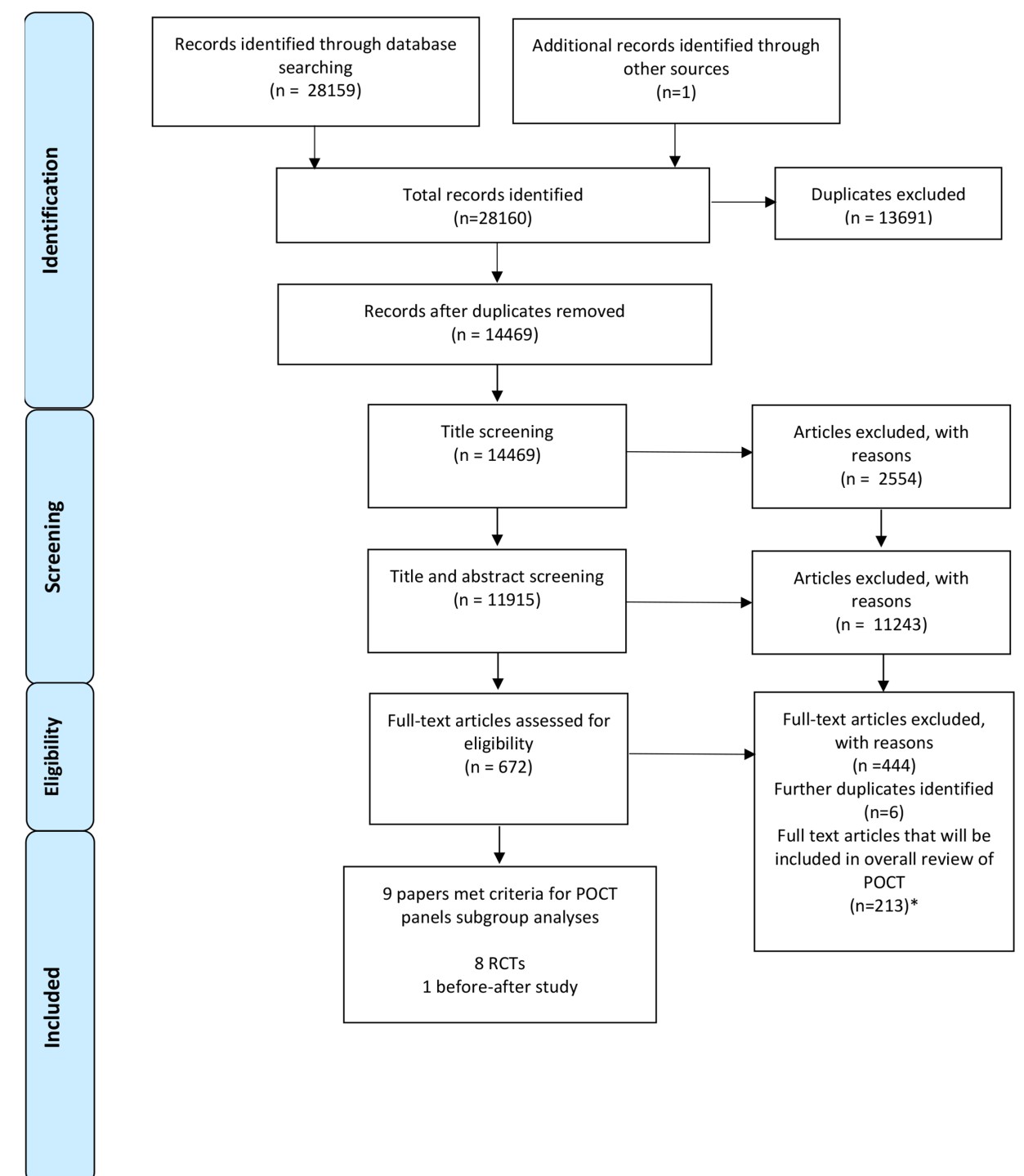

**Figure 1** Preferred Reporting Items for Systematic Reviews and Meta-Analyses flow diagram. *The 22 October 2019 review updates only screened studies for their inclusion in this systematic review, focusing on the impact of point-of-care panel tests (POCT) in ambulatory care, and did not assess suitability for inclusion in the overall review. The 213 articles currently included in the overall POCT review are correct up to the previous update on 17 March 2017.

year, study design and location/setting), inclusion and exclusion criteria and further information regarding the study population (to include mean age and severity of illness of participants). Details of the POCT intervention including which parameters were measured by the POCT device; details of the comparator which was normally conventional blood test were sent to laboratory;

and finally outcomes assessed as listed below were also recorded.

The methodological quality of the included trials was assessed by two authors (AVDB and CG and independently checked by TA). Any areas of conflict were discussed and resolved with a third member of the team where necessary. For RCTs we used the Cochrane Risk of

Bias tool[11] including analysis of randomisation, allocation concealment, comparison of baseline characteristics and blinding. For non-randomised but experimental and controlled studies we used the Cochrane Risk of Bias tool plus an assessment of confounders[15] that were prespecified and included assessing whether baseline characteristics were reported, whether they were similar in intervention and control groups and whether there was a detailed description of the usual care pathway.

### Outcomes assessment

The primary outcome of interest was the impact of POCT on the time to decision regarding disposition, that is, admission/referral termed disposition decision (DD) time. Secondary outcomes included length of stay (LOS) at the ambulatory care unit/practice and mortality. Hospital admission rates, rates of repeat attendance after discharge/readmission were also examined.

### Statistical analyses

Individual study estimates were pooled in a meta-analysis using random-effects inverse-variance model, and study-to-study heterogeneity was assessed using the $I^2$ test statistic in combination with visual inspection using Review Manager.[16] We used mean differences in DD and LOS time (minutes) and their corresponding 95% CIs. Where studies reported the median time to DD and LOS, attempts were made to contact the original authors for mean times and SD. In the case where they were not available, we estimated them using an approach suggested by Wan *et al*[17] which approximates reported medians and quartiles/ranges to corresponding mean and SD robustly by also taking into account studies' sample sizes to avoid small study bias.

### RESULTS

### Description of included studies

The combined total of the original search and update was 28 160 studies as summarised in PRISMA[18] diagram (figure 1). Nine studies relevant to blood-based POCT were selected and reported here, including eight RCTs and one before-after study. Seven studies reported on general panel tests and two studies focused on cardiac panels.

In total, 19 562 participants were included (see study characteristics, table 1). The majority of participants were either adults or defined as being aged over 15 years, with the exception of one study,[19] which recruited from a paediatric emergency departments (EDs) where all participants were aged under 21 years. All the studies were based in EDs except one study[20] that was based in the Canadian ambulance service. Notably, there were no studies based in primary care that focused on panel testing for diagnosis in the acute setting, there were only studies that monitored patients with chronic disease or analysed single tests.

A variety of different POCT panel devices were used in the studies. Although there was variability regarding the specific tests performed by different devices, general panels always included basic metabolic parameters such as sodium, potassium and glucose. Creatinine and basic blood gas analysis such as total carbon dioxide and base excess were also commonly featured. Cardiac panels always included troponin in combination with B-type natriuretic peptide[20] or creatine kinase (myocardial type) and myoglobin.[21]

There was variation in participant inclusion criteria. Two studies included a representative sample of adult ED patients who needed blood tests,[22 23] and one included patients 'whose physicians ordered a comprehensive metabolic panel.'[24] Two studies randomised all patients seen in ED but limited inclusion to the trial to only those patients whose blood work fell entirely within capabilities of the POCT devices used.[19 25] Only one study[26] recorded data on the number of patients who also required tests that were beyond the scope of the POCT device. Personal communication to authors was attempted to obtain this data from the other studies but it had either not been recorded or authors did not respond. Two studies excluded patients who required critical care.[19 24] The cardiac panel studies were more specific in their inclusion criteria, including only patients with chest pain and/or dyspnoea[20] and also had more extensive exclusion criteria such as patients with myocardial infarction on ECG.[21]

Figure 2 summarises the key features of methodological assessment for the eight included RCTs and the risk of bias for the before-after study is available in online supplementary appendix 2. In general, for the included RCTs methodological quality was variable, with the exception of one study[26] being at high risk or unclear for most domains. The before-after study[27] also assessed as 'high' or 'unclear' on all domains; it was also 'high risk' for the confounders assessment as neither the baseline characteristics of participants nor the care pathway for the control group were described in detail.

### PRIMARY OUTCOME

### Disposition decision time

The DD time was specifically reported in three studies.[14 19 26] As summarised in figure 3, POCT reduced the overall DD time by 39 min (95% CI −42.2 to −36.6, $I^2$=0%) compared with usual care. This reduction was increased to 48 min in patients who did not require additional laboratory tests (95% CI −61.11 to −34.05, $I^2$=0%). Hsiao *et al*[19] recruited only from paediatric ED (patients aged under 21 years) while the other studies included adult patients. These all evaluated blood-based panel POCT devices in general ED patients, but Hsiao only reports results for patients whose blood work fell entirely within the capabilities of the POCT device, whereas Illahi *et al*[26] report these different subgroups of patients separately. Illahi *et al*[26] reported point estimates as average values and this has been taken

**Table 1** Characteristics of included studies, setting, device, tests performed, patient characteristics and sample size

| Study | Setting | Device (manufacturer) | Tests performed | Participant characteristics | Sample size |
|---|---|---|---|---|---|
| Parvin et al[27]* | ED USA | i-STAT (i-STAT Corp, Princeton, NJ) | Sodium, potassium, chloride, blood urea nitrogen, glucose, haematocrit, haemoglobin | Patients presenting to the ED between Dec 1994 and Jan 1995 (control period 1), Feb and April 1995 (intervention period) and April 1995 (control period 2) and who had blood tests done that were available on the i-STAT. Only 5.3% of patients had no other central laboratory testing performed in addition to i-STAT | 2067 |
| Kendall et al[22] | ED UK | i-STAT (Abbott) two cartridges evaluated | Sodium, potassium, chloride, urea, glucose, packed cell volume (PCV), calculates Hb from PCV pH, partial pressure carbon dioxide (ppCO$_2$), partial pressure oxygen (ppO$_2$), bicarbonate, total CO$_2$, base excess, oxygen saturation | Representative sample of adult ED patients who needed blood tests. No exclusion criteria. | 1728 |
| Murray et al[25] | ED Canada | NOVA 16 CRT Spectral Cardiac STATus Test Kit (Nova Biomedical) | Creatinine, sodium, potassium, chloride, total CO$_2$, glucose, blood urea nitrogen, haematocrit, qualitative creatine kinase MB isoenzyme (CK-MB), and myoglobin | Adult ED patients, all patients seen in ED randomised but only those patients whose blood work fell entirely within capabilities of POCT were selected. | 180 |
| Hsiao et al[19] | Tertiary paediatric ED USA | i-STAT (Abbott) | Sodium, potassium, chloride, bicarbonate, glucose, blood urea nitrogen, creatinine, ionised calcium, haematocrit, basic blood gas analysis | Paediatric ED patients aged under 21 years old whose blood work fell entirely within capabilities of POCT. Critically ill patients excluded. | 239 |
| Lee et al[23] | Multicentre: 5 EDs South Korea | Piccolo xpress (Abbott) Piccolo Metabolic Reagent Discs | Protein, albumin, alk phos, alanine aminotransferase, aspartate aminotransferase, nitrogen, calcium, cr, glucose, potassium, sodium, bilirubin, total CO$_2$ | ED patients aged 15 years and older clinically required to have chemistry laboratory tests. | 2323 |
| Illahi et al[26] | ED UK | Xpand Plus analyser (Siemens) XS 1000 analyser (Sysmex) | Albumin, Alkaline phosphatase, Amylase, Bilirubin, Calcium, Creatinine, CRP, glucose, paracetamol, phosphate, potassium, sodium, urea, FBC, WBC and differential | Adult ED patients. Some samples then underwent further testing in the central laboratory if required tests were not available from POCT device. | 47 |
| Jang et al[24] | ED Korea | Piccolo xpress (Abbott) Piccolo metabolic reagent discs | Total protein, albumin, alkaline phosphatase, alanine aminotransferase, aspartate aminotransferase, urea, nitrogen, calcium, chloride, creatinine, glucose, potassium, sodium, total bilirubin and total carbon dioxide | ED patients aged 15 years and older whose physicians ordered a comprehensive metabolic panel. Critically ill patients excluded. | 10244 |

Continued

**Table 1** Continued

| Study | Setting | Device (manufacturer) | Tests performed | Participant characteristics | Sample size |
|---|---|---|---|---|---|
| Goodacre[21]† | Multicentre 6 EDs UK | Stratus CS (Siemens) cardiac analyser panel | CK, myocardial type, myoglobin, troponin 1 | Adult ED patients with chest pain. Several exclusion criteria applied including patients with ECG changes consistent with myocardial infarction/high risk acute coronary syndrome, confirmed or suspected serious non-coronary pathology. | 2243 |
| Ezekowitz et al[20]† | Ambulance service Canada (Out of hospital) | Cardio2 panel (Alere) | Troponin and B-type natriuretic peptide | Adults >18 years of age who activated emergency medical services for acute chest discomfort or dyspnoea for which acute cardiovascular disease was deemed to be the most probable diagnosis. Patients excluded if ST-elevation on ECG and non-cardiovascular cause suspected / recurrent dyspnoea. | 491 |

*All studies were RCTs except from Parvin 1996(26) which was a before-after study design.
†Assessed cardiac panels.
ED, emergency department; POCT, point-of-care panel testing.

**Figure 2** Risk of bias summary for included randomised-controlled trials.

as median values in our analysis, attempts were made to contact the author for confirmation but this was not successful. Sensitivity analysis, excluding Illahi,[26] demonstrated robust findings (online supplementary appendix 3). Kendall et al[22] did not specifically measure DD so their results were not included in the meta-analysis. However they did describe that decisions regarding the management plan were made 74 min earlier (95% CI 68 to 80, p<0.0001) when POCT was used for haematological tests as compared with central laboratory testing and 86 min earlier (95% CI 80 to 92, p<0.0001) for biochemical tests.

## SECONDARY OUTCOMES
### Length of stay
LOS in ED was measured in six studies.[19 21 22 24 25 27] Four of these studies[19 22 24 25] were RCTs that assessed general

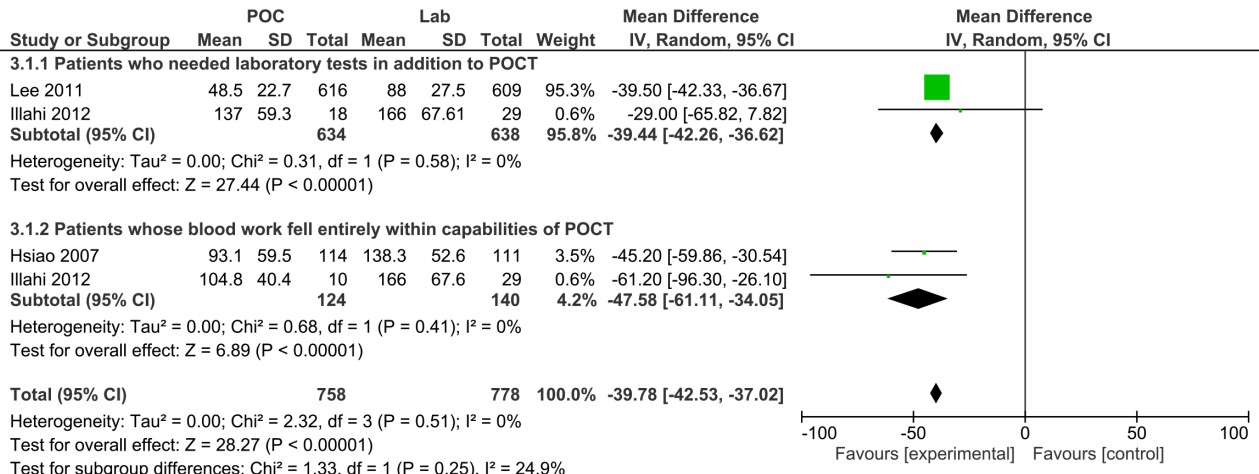

**Figure 3** Forest plot of comparison of time to disposition decision in minutes for patients who needed laboratory testing in addition to point-of-care testing (POCT) and for patients whose blood work fell entirely within the capabilities of POCT.

POCTs in ED and these were combined in the meta-analysis, summarised in figure 4. These included three studies with adult participants[22 24 25] and one study[19] based in paediatric ED. A significant reduction in ED LOS of 33 min (95% CI –60.66 to –5.84) was observed in the POCT group although wide 95% CIs were noted (figure 4). This reduction was increased to 37 min (95% CI –53.08 to –21.77) in patients who only required POCT (and needed no additional laboratory tests). Three of these studies[19 24 25] provided further specific data on the LOS for patients who were admitted and discharged. These data were combined in figure 5, POCT was found to reduce the overall LOS for patients who were later discharged by 34 min (95% CI –63.68 to –5.16) although wide CIs were noted. There was no statistically significant difference between LOS in POCT versus usual care in patients who were later admitted (figure 5). In their before-after study of 4985 patients Parvin et al[27] evaluated a general POCT panel in ED; median LOS with POCT was 209 min (111 to 368) versus 201 (106 to 345) for usual care which was not statistically significant. Subgroup analysis by presenting symptoms and discharge/admit status did not detect any further differences.

LOS in ED was also measured in two studies on POCT cardiac panels.[20 21] One study integrated POCT into emergency medical services in Canada[20] and assessed patients with chest pain or dyspnoea, they found no difference in time from first medical contact to final disposition (9.2 (95% CI 7.3 to 11.1) hours for the POCT group and 8.8 (95% CI 6.3 to 12.1) hours for usual care (p=0.609)). Goodacre et al[21] recorded successful discharge home from ED for patients with chest pain which they defined as having left hospital (or awaiting transport) within 4 hours of arrival and no adverse events occurring over the next 3 months. POCT cardiac biomarker panels were associated with an increased rate of successful discharge (32% vs 13% in the usual care group, OR 3.81, 95% CI 3.01 to 4.82; p<0.001), although analysis of the original data demonstrated that the median LOS in ED for the POCT group was longer at 216 min interquartile range (IQR) 179–238) compared with the usual care pathway of 188 min (IQR 142–225).

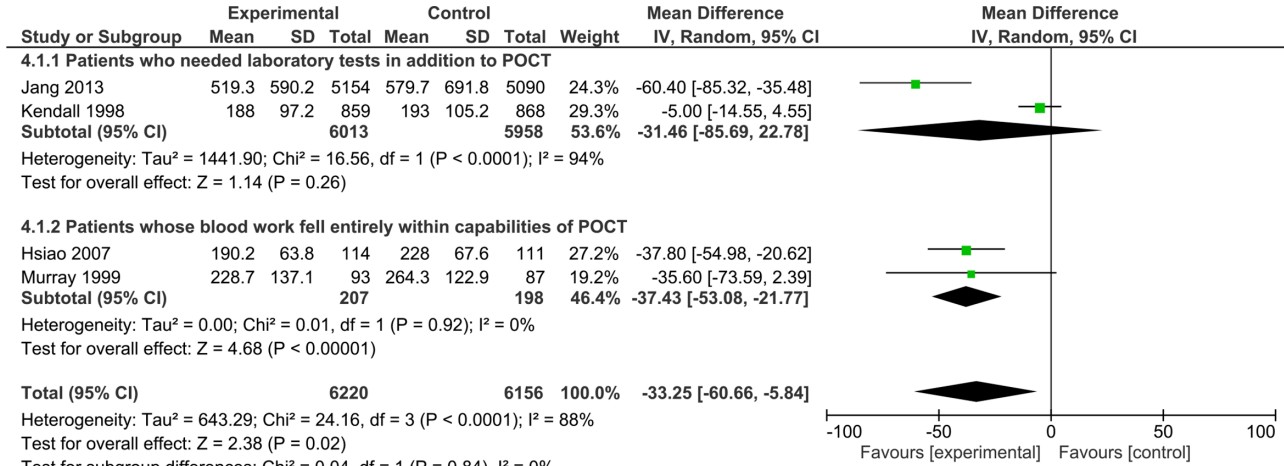

**Figure 4** Forest plot of comparison of length of stay time in minutes for patients who needed laboratory testing in addition to point-of-care testing (POCT) and for patients whose blood work fell entirely within the capabilities of POCT.

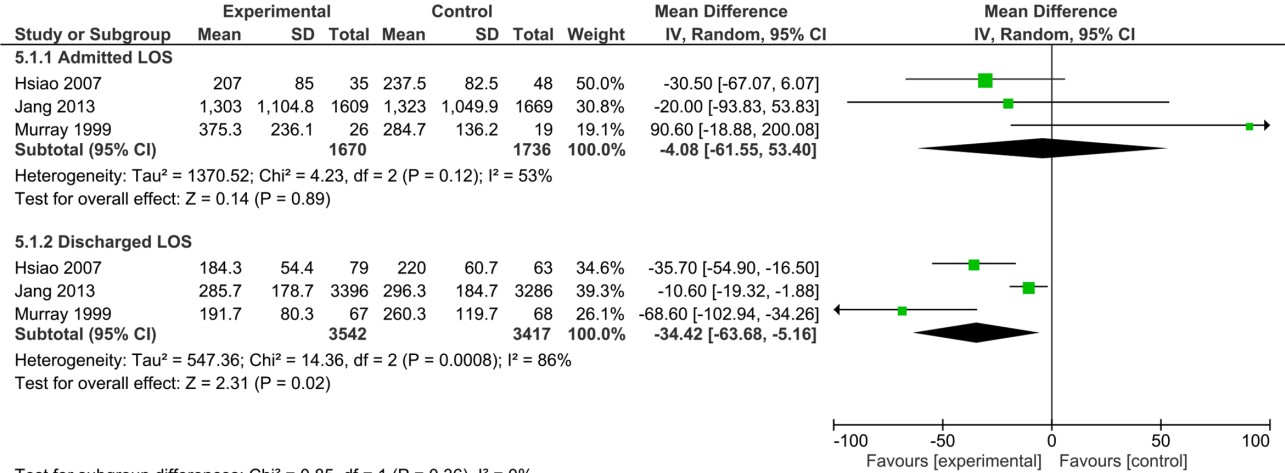

**Figure 5** Forest plot of comparison of length of stay time in minutes for patients who underwent point-of-care testing versus laboratory testing, split into subgroups for patients who were admitted/discharged.

### Mortality

Three studies included data on patient mortality.[20–22] There was no significant difference in mortality between POCT and laboratory testing as demonstrated in figure 6. Two of these studies evaluated cardiac panels, calculated risk ratios of death were 2.98 (0.60 to 14.74)[21] and 0.80 (0.22 to 2.94),[20] one study on general panels reported a relative risk of death of 1.16 (0.79 to 1.68).[22]

For the other secondary outcomes, only one study reported hospital admission rates[28] and found that this was not significantly different between the POCT and laboratory groups (difference 1.7, CI –1.7 to 5.1, p=0.33). Rates of repeat ED attendance after discharge and rehospitalisation were also recorded by Ezekowitz et al[20] and there was no significant difference detected between POCT and laboratory testing (p=0.320, p=0.712, respectively).

In terms of exploratory outcomes, there is evidence that unwell patients benefited from faster decision making with POCT. Kendall et al[22] describe how 59 out of 859 POCT patients had changes in their management in which timing was considered to be critical; these included decision to intubate/ventilate. POCT was also associated with reduced time to CT from ED arrival,[24] with a median difference of 11 min (95% CI 3 to 19).

### DISCUSSION
#### Statement of principal findings

This systematic review found that general panel tests performed at the point-of-care may result in faster disposition and management decisions, which in turn might reduce LOS for patients who are subsequently discharged from the ED. This is not associated with changes in mortality. There is also no gain in LOS for patients who are admitted to hospital. These results perhaps suggest that specific groups of patients may benefit from the introduction of POCT in an ED setting such as, well patients who could be discharged faster and unwell patients who need critical interventions more quickly. The LOS advantage was attenuated when extra tests were required from the laboratory in addition to the POCT panel.

#### Strengths and weaknesses of the study

This study was conducted robustly, using a comprehensive search strategy in the major medical databases, and selection and quality assessment performed by two independent reviewers. It offers results on the impact of panel POCT, rather than just their accuracy. A comprehensive search strategy also brings limitations, variation in countries, healthcare practices and usual care may have

**Figure 6** Relative risk of death in POCT compared with laboratory testing for general and cardiac panel tests performed in ED, and cardiac panels tested by the ambulance service. ED, emergency department; POCT, point-of-care testing.

contributed to high heterogeneity for some outcomes. Another consideration is that we present here studies published from 1996 to 2014 and clinical practice will have changed during this time, moreover it is concerning that no new impact evaluations of blood-based panels have been performed in the last 5 years despite the increase in implementation of POCT. Although impact studies are an integral part of the evidence cycle for new tests,[28] they are also difficult to organise and subject to bias. In our review, blinding clinicians and patients from the intervention was not possible by nature, introducing a risk of bias and in general most studies were not blinded by outcome assessment. Only a small number of relatively small studies were included in this meta-analysis which did not allow further exploration of small study effect. Moreover, most notably for mortality, the results may suffer from being underpowered to detect differences between POCT and laboratory testing.

Statistical and clinical heterogeneity is evident within our meta-analysis, particularly for LOS results. The meta-analysis was considered carefully and reduced where possible by ensuring that studies of cardiac and general panels tests were not combined; moreover, we did not combine the before-after study (which also had a high risk of bias)[27] with RCTs. Multiple factors influence our primary and secondary outcomes and these variables are responsible for much of the clinical heterogeneity. It is important to consider the system in which POCT is implemented and which ED triage systems are used. For example, if blood tests are requested on arrival in ED than laboratory results might be available at the time of physician review anyway and thus there would be fewer benefits to POCT. Moreover, practicalities such as how quickly radiology is available and how samples are transported to laboratories will impact results significantly.

There are also many factors which impact ED LOS specifically, especially availability of inpatient beds and this may be the reason for the reduced benefit on LOS in the admitted group. Other important factors that differed between the studies and between different hospitals,[21] included the time of day that POCT was available; with one study only performing POCT during working hours, as well as the availability and seniority of clinical staff. Furthermore, the studies differed in their inclusion criteria, and as demonstrated by our subgroup analysis, the benefits of POCT on LOS were proportional to the spectrum of tests available. This perhaps explains why Parvin et al[27] did not demonstrate any benefit in reduced LOS from POCT as 95% of these patients also required additional laboratory tests in addition to the POCT panel.[28] Moreover, there is further evidence of this association from other studies that combined single and multiple tests and demonstrated a significant reduction in LOS for POCT.[29 30]

Other factors relate specifically to study protocol, for example Goodacre et al[21] describe how the LOS was longer for the POCT because POCT patients did not leave the ED until their POCT testing (at baseline and

90 min) was complete, whereas the standard care group could leave the ED as soon as medical assessment was complete and a decision to admit (or discharge in a few cases) had been made. Therefore, the POCT group spent longer in the ED but were more likely to go home before the 4-hour point, while the usual care group spent less time in the ED because they were more likely to be admitted to a ward (and thus leave the ED) at any earlier time (personal communication with author).

An important limitation to highlight is that four studies excluded critically ill patients[19 24] and patients with myocardial infarction,[20 21] which may have biased mortality data.

## Comparison with other studies

The benefit of POCT in ambulatory patients has been shown by Kankaanpää et al,[31] where single and panel tests were implemented in ambulatory patients presenting to a Finnish ED who also see primary care patients outside of office hours. They excluded all patients who were admitted to hospital. Median LOS in the control phase was 3.51 hours (3.38–4.04) and this was reduced to 3.22 hours (3.12–3.31, p=0.000) with the implementation of POCT; moreover, the combination of POCT with an early assessment triage model, reduced LOS to 3.05 hours (02.59–03.12, p=0.033). This study[31] appropriately recorded which patients also required additional laboratory testing and found that this was lowest when POCT and an early assessment triage model were combined, when 68% of patients did not require additional blood tests (which was also associated with the greatest reduction in LOS).

Lingervelder et al[32] performed a systematic review to assess POCT implementation aspects addressed in primary care. They found that only 8% of evaluations included measurement of clinical utility, even though GPs perceive this as the most important issue to consider. They found that the most frequently evaluated tests were single tests such as HbA1c, CRP and D-dimer so these would not have been included in our panels review.

## Implications for research and practice

Future research is required to understand the impact that POCT panels have in assisting with the decision to admit or discharge patients and analyse their cost-effectiveness.[33 34] We would recommend that future trials assess successful discharge, rate of admission and rate of adverse events rather than just focusing on time to discharge or DD. Moreover, the relationship between ED overcrowding and LOS needs to be better understood as reductions in LOS do not necessarily reduce overcrowding.[35] This review suggests that there are specific subgroups that may benefit most from the implementation of POCT, and future studies should focus on these groups and establish which tests should be combined in a POCT panel such as CRP.

Theoretically there are also advantages to using POCT in the primary care setting. For example, it may help to identify acute kidney injury or atypical presentation of

myocardial infarction. However, it may not be time efficient or cost-effective. As none of the included studies were based in primary care, understanding the impact of POCT in this setting remains a research priority. Research outcomes and study designs in this environment need to be carefully considered, particularly as laboratory testing may not be available at all or maybe delayed, particularly regarding home visits and for patients in rural areas.

It is important to understand how POCT changes management decisions particularly regarding admission and to monitor whether the thresholds for ordering tests changes with POCT implementation.[34] Future research should also consider patients views on POCT and how implementation could be linked to digital transformations to maximise benefits for patients and clinicians alike.

There is a clear gap between evidence and policy which needs to be addressed. NHS England have stated that the POCT i-STAT will be available in urgent treatment centres in the UK from 2019[6] but there is currently a lack of evidence to underpin this. Future research needs to based in ambulatory settings and should identify how POCT can contribute meaningful changes to patient care rather than simply examining healthcare processes and must focus on the impact of POCT implementation.

Copyright

**Acknowledgements** We thank Nia Roberts, Dr Jennifer Hirst and Dr José M. Ordóñez-Mena for their assistance with this project.

**Contributors** TA designed the initial search strategy. Screening was carried out by JV, JJL, CG, PST, TA, PJT and AVDB. Data extraction was performed by CG, AV and TA. CG and PST carried out the meta-analysis. CG and AV drafted the manuscript. All authors commented and co-drafted the final version of this article. CG is the guarantor and affirms that the manuscript is an honest, accurate and transparent account of the study being reported; that no important aspects of the study have been omitted and that any discrepancies from the study as planned (and, if relevant, registered) have been explained.

**Funding** This paper presents independent research funded by the National Institute for Health Research School for Primary Care Research (NIHR SPCR), funding round 11, award number 309. CG is a Wellcome Trust Doctoral Fellow [grant number 203921]. JV, PT, GH and AV are supported through the National Institute for Health Research (NIHR) Community Healthcare MedTech and IVD Co-operative Oxford at Oxford Health Foundation Trust (award number MIC-2016-018). JL is an NIHR In-Practice Fellow.

**ORCID iDs**
Clare Goyder http://orcid.org/0000-0002-7461-9661
Gail Hayward http://orcid.org/0000-0003-0852-627X

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
