## [Reviewer comments · BMJ Open]

ARTICLE DETAILS

TITLE (PROVISIONAL)	Impact of point-of-care panel tests in ambulatory care: a systematic review and meta-analysis
AUTHORS	Goyder, Clare; Tan, Pui San; Verbakel, Jan; Ananthakumar, Thanusha; Lee, Joseph J; Hayward, Gail; Turner, Philip J; Van Den Bruel, Ann

VERSION 1 – REVIEW

REVIEWER	Lesley Scott, Head Research and Development, Associate Professor Department of Molecular Medicine and Haematology, University of the Witwatersrand, School of Pathology, Faculty of Health Science, Johannesburg, South Africa
REVIEW RETURNED	15-Aug-2019

GENERAL COMMENTS	Review: Impact of Point-of-care panel tests in ambulatory care: a systematic review and meta-analysis. General: The body of work presented is extensive and contributes unique knowledge to the field and is well summarised. Attention to detail is required for figure numbers and overall abbreviations are poor and difficult for readers not skilled in the terms to follow. All Figure legends require explanation, and Tables given are poorly presented, as if they are extracted from a statistical output. Only 1 or 3 decimals is required and current presentation is hard to follow. Some corrections noted: Importance: line 26 – “the” is incorrect. Reference 10 should be updated? Selection of studies: line 21: before and after, comparing Abstract:  • Please provide a definition or a different term for usual care group – perhaps standard of care • The last sentence (lines 32-36) could be re-worded: requires clarification on health care processes – in regards to primary health care. This sentence is really a discussion point, not the main findings from this study, and misleading to readers who may perceive the meta-analysis to have more details on primary care POCT. Strengths and limitations:  • Bullet 3: line 11 requires better wording (more scientific) for “power problems”- if so then provide recommendations or give further reasons Outcomes assessment:  • Better define healthcare processes Results:
---

	 • Table 1: please add a column after the test measure to include time (could be a range) to reported result for each analyte off the POCT. Discussion:  • Please add discussion on: Possible limitations could include the time period between the various studies 1996-2014. Would be good to include if this could have changed POCT technology or practices. The studies come from vastly different country settings, would standard of care (usual) perhaps impact on some of the study outcomes. • Weaknesses: Readers may find it more valuable to represent the identified weaknesses in a table listed more as considerations for POCT (lines 44 – next page 7), which could be turned into guidance for future studies and future implementation partners. • Implications for research and practice: line 2 pg 14. It is unclear what is meant by “consider the patient’s perspective”. This could be expanded, and perhaps linked to the digital transformation era and further discussed.
--	--

REVIEWER	Nathorn Chaiyakunapruk University of Utah, USA
REVIEW RETURNED	16-Sep-2019

GENERAL COMMENTS	The authors performed appropriate statistical analyses in this study. The authors could have mentioned the limitation of the small number of studies included in this review which did not allow them to explore small study effect. The authors acknowledged the evident heterogeneity and tried to minimize unnecessary pooling of the studies.
---

REVIEWER	Michelle M.A. Kip University of Twente, the Netherlands
REVIEW RETURNED	04-Oct-2019

GENERAL COMMENTS	Thank you for the opportunity to review this very interesting and highly relevant manuscript. The manuscript provides a very complete and extensive overview of the impact of the use of POCT panels in ambulatory care. The systematic review and meta-analysis have been carried out very accurately and the methods and results are described in detail. I have two minor comments to take into consideration:  - During the screening process, 2554, 9715 and 428 articles were excluded. The manuscript would benefit from a description (or summary) of why these articles were excluded. It is currently stated in one paragraph that 26,124 articles were found, resulting in 225 papers in the overall review, and the inclusion of only 9 relevant studies. Although it is described in the 'selection of studies' paragraph why they were excluded, it is currently not clear what were the most common reasons for exclusion, and what exclusion criteria were applied in which step of the reviewing process. - Of the 9 articles that were included, 3 were published in 1996 or 1998. As the diagnostic performance of POCTs has improved over the last decades, I find it hard to judge whether the results from these 3 papers are still applicable nowadays or whether they may actually be outdated. Therefore, please consider to address this issue in the discussion section of the manuscript.
---

VERSION 1 – AUTHOR RESPONSE

Reviewer: 1

Reviewer Name: Lesley Scott

Comment 8

General: The body of work presented is extensive and contributes unique knowledge to the field and is well summarised. Attention to detail is required for figure numbers and overall abbreviations are poor and difficult for readers not skilled in the terms to follow.

Response to comment 8

Thank you very much for this helpful feedback. We have revised the document to ensure that figure numbers are correct and to ensure clarity and consistency in abbreviations used.

Comment 9

All Figure legends require explanation, and Tables given are poorly presented, as if they are extracted from a statistical output. Only 1 or 3 decimals is required and current presentation is hard to follow.

Response to comment 9.

Thank you. We have added figure legends (at the end of the document before references) and improved presentation of tables. Table 1 has been constructed in word. We have changed all numbers to 1 decimal point format.

Comment 10

Some corrections noted:

Importance: line 26 – “the” is incorrect. Reference 10 should be updated?

Response to comment 10

Thank you we have removed “the” from this line. We have kept the 2004 reference but have added a 2019 systematic review into our discussion paragraph.

Comment 11

Selection of studies: line 21: before and after, comparing

Response to comment 11

Thank you – we have changed this to before-after studies which is consistent with previous terminology used.

Comment 12

Abstract:

Please provide a definition or a different term for usual care group – perhaps standard of care

Response to comment 12

Thank you for raising this point. The usual care group are patients that received laboratory testing instead of POCT. The abstract has now been edited to reflect so we hope this is now clearer.

Comment 13

The last sentence (lines 32-36) could be re-worded: requires clarification on health care processes – in regards to primary health care. This sentence is really a discussion point, not the main findings from this study, and misleading to readers who may perceive the meta-analysis to have more details on primary care POCT.

Response to comment 13

Thank you for this comment. We have restructured the abstract and made it clear in the results section that no studies from primary care settings were identified.

Comment 14

Strengths and limitations:

Bullet 3: line 11 requires better wording (more scientific) for “power problems” - if so then provide recommendations or give further reasons

Response to comment 14

Thank you for this feedback – we have changed the wording here to clarify this point

Comment 15

Outcomes assessment:

Better define healthcare processes

Response to comment 15

Thank you for highlighting this, we now given examples of healthcare processes to explain this term more clearly to the reader.

Comment 16

Results:

Table 1: please add a column after the test measure to include time (could be a range) to reported result for each analyte off the POCT.

Response to comment 16

Thank you for this suggestion. We were not able to fit this into the table but have included information on time taken to do tests in the 2nd paragraph of the background section.

Comment 17

Discussion: Please add discussion on: Possible limitations could include the time period between the various studies 1996-2014. Would be good to include if this could have changed POCT technology or practices. The studies come from vastly different country settings, would standard of care (usual) perhaps impact on some of the study outcomes.

Response to comment 17

Thank you for bringing this to our attention. We have added information on this point into paragraph 2 of the discussion, strengths and weaknesses of the study section.

Comment 18

Weaknesses: Readers may find it more valuable to represent the identified weaknesses in a table listed more as considerations for POCT (lines 44 – next page 7), which could be turned into guidance for future studies and future implementation partners.

Response to comment 18

Thank you for this suggestion. We preferred to keep this information in the discussion which covers implementation, rather than put it in a table.

Comment 19

Implications for research and practice: line 2 pg 14. It is unclear what is meant by “consider the patient’s perspective”. This could be expanded, and perhaps linked to the digital transformation era and further discussed.

Response to comment 19

Thank you for suggesting this change. We have now edited the final discussion paragraph to include the following: “Any future research should identify how POCT can contribute meaningful changes to patient care rather than simply look at health care processes, should consider what approaches are most favoured by patients and how implementation could be linked to digital transformations to maximise the benefits.”

Reviewer: 2

Reviewer Name: Nathorn Chaiyakunapruk

Comment 20

The authors performed appropriate statistical analyses in this study. The authors could have mentioned the limitation of the small number of studies included in this review which did not allow them to explore small study effect. The authors acknowledged the evident heterogeneity and tried to minimize unnecessary pooling of the studies.

Response to comment 20

Thank you for this important observation. We have now included this in the strengths and weaknesses paragraph in the discussion.

Reviewer: 3

Reviewer Name: Michelle M.A. Kip

Comment 21

Thank you for the opportunity to review this very interesting and highly relevant manuscript. The manuscript provides a very complete and extensive overview of the impact of the use of POCT panels in ambulatory care. The systematic review and meta-analysis have been carried out very accurately and the methods and results are described in detail.

Response to comment 21

Thank you very much for this positive comment, we are glad that this work was interesting and highly relevant.

Comment 22

During the screening process, 2554, 9715 and 428 articles were excluded. The manuscript would benefit from a description (or summary) of why these articles were excluded. It is currently stated in one paragraph that 26,124 articles were found, resulting in 225 papers in the overall review, and the inclusion of only 9 relevant studies. Although it is described in the 'selection of studies' paragraph why they were excluded, it is currently not clear what were the most common reasons for exclusion, and what exclusion criteria were applied in which step of the reviewing process.

Response to comment 22

Thank you very much for these comments. We have now revised the PRISMA diagram so it accurately reflects this specific review process for this review and we have updated the 'selection of studies' paragraph to explain the common reasons for exclusion.

Comment 23

Of the 9 articles that were included, 3 were published in 1996 or 1998. As the diagnostic performance of POCTs has improved over the last decades, I find it hard to judge whether the results from these 3 papers are still applicable nowadays or whether they may actually be outdated. Therefore, please consider to address this issue in the discussion section of the manuscript.

Response to comment 23

Thank you for raising this point. We agree that this is an important consideration and have revised paragraph 2 of discussion, strengths and weaknesses of the study, accordingly.

VERSION 2 – REVIEW

REVIEWER	Michelle M.A. Kip University of Twente, the Netherlands
REVIEW RETURNED	03-Dec-2019
GENERAL COMMENTS	Thank you for addressing my comments.